# A crowdsourced set of curated structural variants for the human genome

**Lesley M. Chapman**[1¤a], **Noah Spies**[1,2,3¤b], **Patrick Pai**[4], **Chun Shen Lim**[5], **Andrew Carroll**[6], **Giuseppe Narzisi**[7], **Christopher M. Watson**[8,9], **Christos Proukakis**[10], **Wayne E. Clarke**[7], **Naoki Nariai**[11], **Eric Dawson**[12,13], **Garan Jones**[14], **Daniel Blankenberg**[15], **Christian Brueffer**[16], **Chunlin Xiao**[17], **Sree Rohit Raj Kolora**[18,19,20], **Noah Alexander**[21], **Paul Wolujewicz**[22], **Azza E. Ahmed**[23], **Graeme Smith**[24], **Saadlee Shehreen**[25], **Aaron M. Wenger**[26], **Marc Salit**[1,2], **Justin M. Zook**[1]\*

1 Biosystems and Biomaterials Division, Material Measurement Laboratory, National Institute of Standards and Technology, Gaithersburg, Maryland, United States of America, 2 The Joint Initiative for Metrology in Biology, Stanford University, Stanford, California, United States of America, 3 Departments of Genetics and Pathology, Stanford University, Stanford, California, United States of America, 4 University of Maryland - College Park, College Park, Maryland, United States of America, 5 Department of Biochemistry, School of Biomedical Sciences, University of Otago, Dunedin, New Zealand, 6 DNAnexus Inc, Mountain View, California, United States of America, 7 New York Genome Center, New York, New York, United States of America, 8 School of Medicine, University of Leeds, Saint James's University Hospital, Leeds, Leeds, United Kingdom, 9 Yorkshire Regional Genetics Service, The Leeds Teaching Hospitals NHS Trust, Saint James's University Hospital, Leeds, United Kingdom, 10 University College London, Institute of Neurology, London, United Kingdom, 11 Illumina, Inc. San Diego, California, United States of America, 12 Division of Cancer Epidemiology and Genetics, National Cancer Institute, National Institutes of Health, Rockville, Maryland, United States of America, 13 Department of Genetics, University of Cambridge, Cambridge, United Kingdom, 14 University of Exeter Medical School, Epidemiology and Public Health Group, Barrack Road, Exeter, Devon, United Kingdom, 15 Genomic Medicine Institute Lerner Research Institute Cleveland Clinic, Cleveland, Ohio, United States of America, 16 Division of Oncology and Pathology, Department of Clinical Sciences Lund, Lund University, Lund, Sweden, 17 National Center for Biotechnology Information, National Library of Medicine, National Institutes of Health, Bethesda, Maryland, United States of America, 18 German Centre for Integrative Biodiversity Research (iDiv) Halle-Jena-Leipzig, Leipzig, Germany, 19 Bioinformatics Group, Department of Computer Science, and Interdisciplinary Center for Bioinformatics, Universität Leipzig, Leipzig, Germany, 20 Molecular Evolution and Systematics of Animals, Institute of Biology, University of Leipzig, Leipzig, Germany, 21 Molecular Biology Institute, University of California Los Angeles, Los Angeles, California, United States of America, 22 Weill Cornell, Belfer Research Building, New York, New York, United States of America, 23 Center for Bioinformatics and Systems Biology, Faculty of Science, University of Khartoum and Department of Electrical and Electronic Engineering, Faculty of Engineering, University of Khartoum, Khartoum, Sudan, 24 Guy's Hospital and St Thomas's NHS Foundation Trust Great Maze Pond, London, United Kingdom, 25 Department of Genetic Engineering & Biotechnology, University of Dhaka, Bangladesh, 26 Pacific Biosciences, Menlo Park, California, United States of America

¤a Current address: National Cancer Institute, iCURE Postdoctoral Fellow, Bethesda, Maryland, United States of America
¤b Current address: Celsius Therapeutics, Cambridge, Massachusetts, United States of America
\* justin.zook@nist.gov

## Abstract

A high quality benchmark for small variants encompassing 88 to 90% of the reference genome has been developed for seven Genome in a Bottle (GIAB) reference samples. However a reliable benchmark for large indels and structural variants (SVs) is more challenging. In this study, we manually curated 1235 SVs, which can ultimately be used to evaluate SV callers or train machine learning models. We developed a crowdsourcing app—SVCurator —to help GIAB curators manually review large indels and SVs within the human genome,

---

---

**Data Availability Statement:** All sequencing data files are available from the SRA database (accession number PRJNA200694)

**Funding:** LMC and JMZ were supported by an interagency agreement with the Food and Drug

Administration (www.fda.gov) and the intramural research program of the National Institute of Standards and Technology (www.nist.gov). CX was supported by the Intramural Research Program of the National Library of Medicine, National Institutes of Health (www.nih.gov). The funders had no role in study design, data collection and analysis, decision to publish, or preparation of the manuscript.

**Competing interests:** AC is an employee of Google Inc. AC is a former employee of DNAnexus Inc. NN is an employee of Illumina Inc.

and report their genotype and size accuracy. SVCurator displays images from short, long, and linked read sequencing data from the GIAB Ashkenazi Jewish Trio son [NIST RM 8391/HG002]. We asked curators to assign labels describing SV type (deletion or insertion), size accuracy, and genotype for 1235 putative insertions and deletions sampled from different size bins between 20 and 892,149 bp. 'Expert' curators were 93% concordant with each other, and 37 of the 61 curators had at least 78% concordance with a set of 'expert' curators. The curators were least concordant for complex SVs and SVs that had inaccurate breakpoints or size predictions. After filtering events with low concordance among curators, we produced high confidence labels for 935 events. The SVCurator crowdsourced labels were 94.5% concordant with the heuristic-based draft benchmark SV callset from GIAB. We found that curators can successfully evaluate putative SVs when given evidence from multiple sequencing technologies.

## Author summary

Large genomic changes, called structural variants, can cause a variety of human diseases, but have been challenging to detect with conventional DNA sequencing methods. We are working in the Genome in a Bottle Consortium to develop authoritatively characterized genomes with benchmark structural variants that can be used by anyone to assess the accuracy of their sequencing and analysis methods. Manual curation of the sequencing reads from multiple technologies has been essential to establish benchmark variant calls. Here, we present consensus curations from a web-based platform that displays a comprehensive set of visualizations of sequencing read support for structural variants. We use the svviz visualization tool to present evidence not only for deletions but also for insertions, which have previously not been possible to curate. We derive consensus calls from the multiple curations of each variant, and we find these are highly concordant with a draft Genome in a Bottle structural variant benchmark set.

## Introduction

Structural variants (SVs) are typically defined as DNA variants $\geq$ 50 base pairs (bp), and include: insertions, deletions, duplications, and inversions[1,2]. SVs have been linked to a number of human diseases[3]. Recent next generation sequencing technologies and analysis algorithms have substantially improved the discovery of SVs. However, identifying SVs with high confidence remains a challenge as evidenced by inconsistent predictions of SVs across different methods[4,5]. Several groups have demonstrated that crowdsourcing applications can be effective for generating labeled data for putative SVs. Greenside et al. used crowdsourcing to label 1781 deletions for the Personal Genomes Project Ashkenazi Jewish Trio son [HG002[4]. Recently, *SV-Plaudit* was used to evaluate 1350 SVs (97% deletions), and allowed participants to evaluate candidate SVs using samplot, which displays images representing short and long read sequencing technologies[6,7]. The web-based platform, Plotcritic, renders samplot images and provides users with an interface to evaluate putative SVs[6].

In the current study, we generated a list of SVs with consensus curation results for SV type, size, and genotype, labels that could be used to train machine learning models to characterize SVs or evaluate SV calling performance. These data were generated via a Python web

application (app)—SVCurator—that enables users to evaluate large indels and SVs from the one human's genome—the GIAB Ashkenazi Jewish Trio son [NIST RM 8391/HG002]. The platform provides a variety of IGV and svviz2 images from short, long, and linked read sequencing data for putative SVs randomly sampled from candidate calls. These SVs were generated from over 30 variant callers using data produced from five different sequencing technologies. To evaluate the accuracy of curations, we discuss the levels of concordance with heuristic-based labels assigned to events within the GIAB v0.6 benchmark SV calls for HG002 [8].

## Results

### SVCurator platform overview

SVCurator is a Python web platform (Fig 1) we developed to evaluate putative large indels ≥20bp and SVs from the union of callsets from diverse technologies and calling methods for the Genome in a Bottle (GIAB) Ashkenazi Jewish Trio son (HG002/NA24385) [ftp://ftp-trace.ncbi.nlm.nih.gov/ReferenceSamples/giab/data/AshkenaziTrio/analysis/NIST_UnionSVs_12122017/SVmerge121217/].

Curators evaluated 1295 SV calls (579 deletions and 716 insertions) that were randomly selected in 7 size bins from a pool of candidate variants (Fig 2). For each SV, SVCurator displays a number of images developed and recommended by experts from the GIAB consortium. Extensive data was generated from short, long, and linked-read whole genome sequencing technologies by the GIAB consortium. These data include Illumina 250bp paired end sequencing, Illumina 150bp paired end sequencing, Illumina 6kb mate-pair, haplotype-partitioned PacBio and haplotype-partitioned 10x Genomics (S1 Fig) [9]. Svviz2 [10] was used to generate images of reads from each dataset aligned to the reference or alternate alleles. Svviz2 was also used to generate dotplots to visualize repetitive regions in the reference and alternate haplotypes and alignments of individual reads to the haplotypes. Images of Illumina 250bp paired end sequencing, haplotype-partitioned PacBio and haplotype-partitioned 10x Genomics in Integrative Genomics Viewer (IGV) were also included [10]. Participants were asked to evaluate each call and determine whether a SV exists at each site within 20% of the called size of the variant, assign a label describing the variant genotype ["Homozygous Reference", "Heterozygous Variant", "Homozygous Variant", "Complex or difficult"] and a confidence score for the variant genotype (GT).

### Selecting top curators that have high concordance with expert curators

To assess the reliability of each curator's results, we evaluated concordance between each curator and a set of 7 known "expert curators." 136 participants registered to use the app, of which 61 evaluated events. Of the 1295 events, 1290 events were curated at least 3 times (S2 Fig), with a mean of 11 curators evaluating each event. The average time to curate each event was 47.31 seconds, though curators' average curation times varied from <10 seconds to >120 seconds (S3 Fig). Labels assigned by each curator were compared to labels assigned by a set of 7 'expert' curators from the GIAB Analysis Team who had experience curating SVs. The expert consensus label was assigned to each event by simple voting (i.e., from the label assigned by the most 'expert' curators). The percent concordance was defined as the fraction of 'expert' curators who agreed on the consensus label divided by the total number of expert curators who evaluated the event. On average, the 'expert' curators were 93% concordant on the labels assigned to each event. Each 'expert' was assigned a percent concordance score based on the level of concordance between their assigned label and the consensus label from the remaining experts.

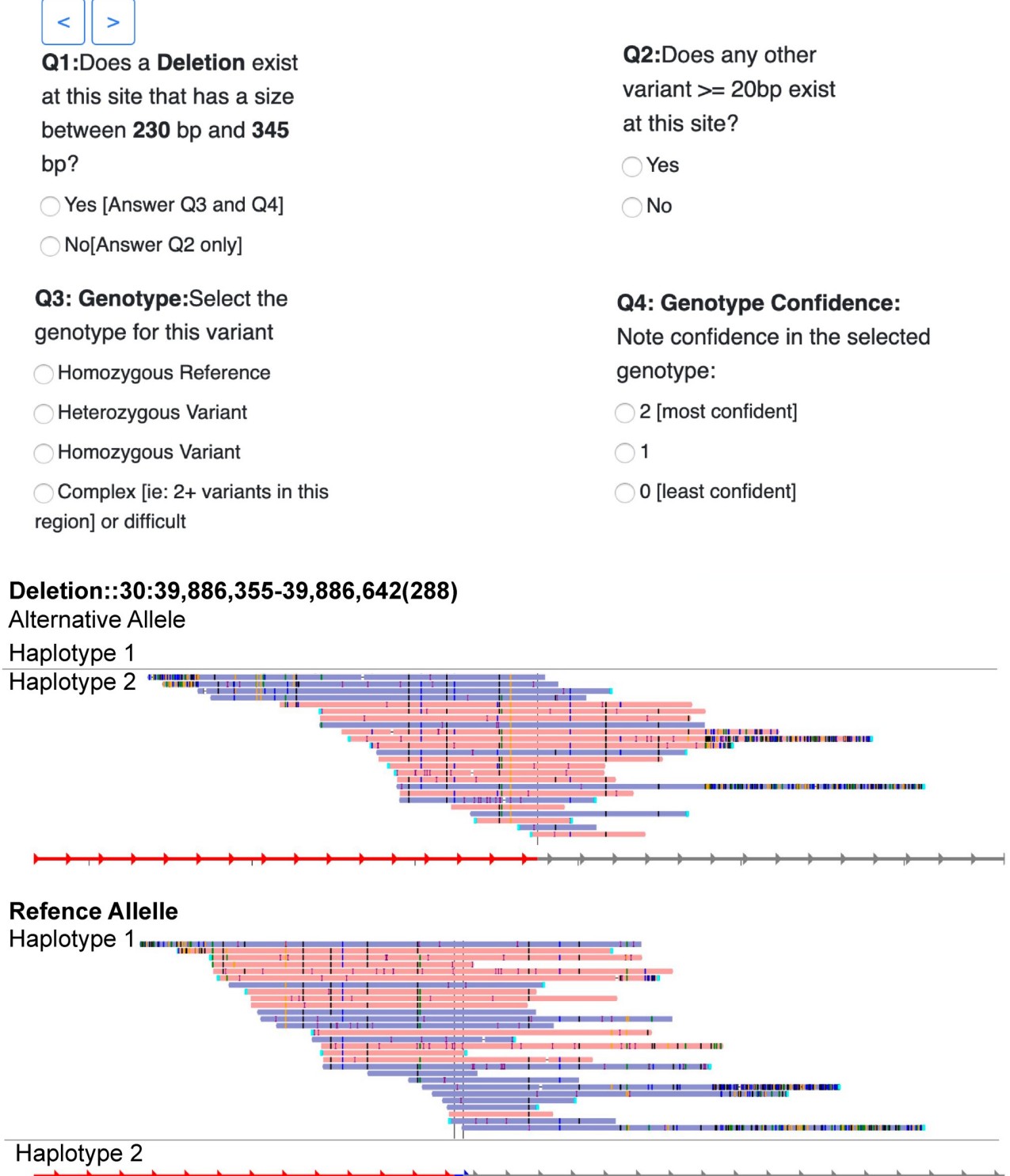

**Fig 1. SVCurator web application interface.** A) 13 images with reads aligned to reference or alternate alleles, as well as dotplot images from svviz2 for several technologies were also visible in addition to the IGV image shown here. B) Curators were asked whether a variant exists within ±20% of the predicted size, its genotype, and their confidence in the curated genotype. C) Example svviz2 haplotype-partitioned image with haplotype 1 reads aligned to the reference and haplotype 2 reads aligned to the alternate allele for a heterozygous 307 bp insertion.

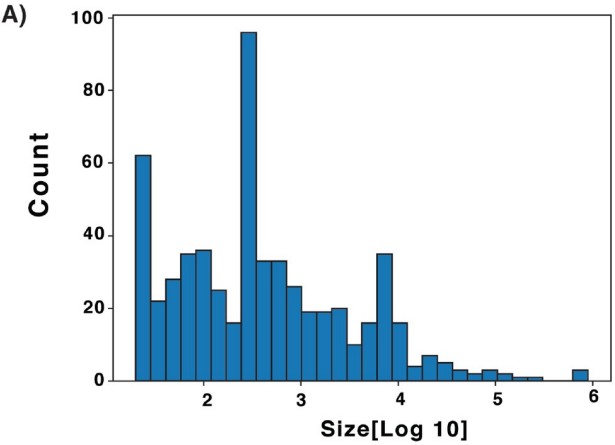
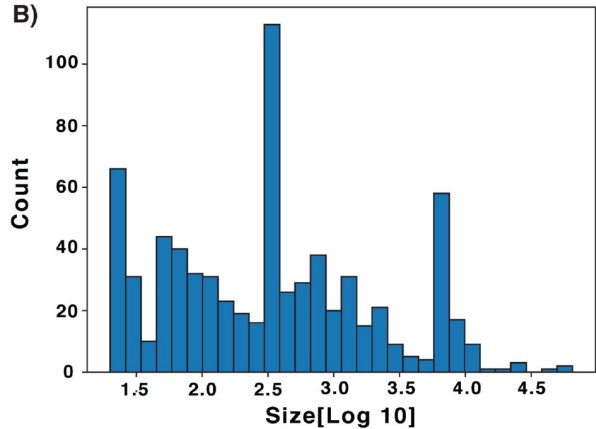

**Fig 2. Events displayed in SVCurator randomly sampled from the GIAB union callset in 7 different size ranges.** A) 579 deletions and B) 716 insertions.

The concordance of each expert with the consensus expert label ranged from 77.7% to 100%. The 7 'expert' curators were 100% concordant for 407 events. Overall, deletions averaged 86% concordance among 'experts' and insertions averaged 80% concordance. There were 298 deletions and 243 insertions where 'expert' curators had at least 68% concordance on the assigned label with 3 or more 'expert' curators who agreed on the assigned label. These 541 events were used to identify 'top' curators from the full set of curators.

There were 20 curators (including 5 'expert' curators) who evaluated more than 648 SVCurator events. Of these, a mean of 670 events per curator were used in further analysis after filtering responses where participants were unsure about an event existing at a particular site or assigned low genotype confidence scores [Genotype confidence score = 0]. These 20 curators had on average 87% concordance with 'expert' consensus labels (Fig 3).

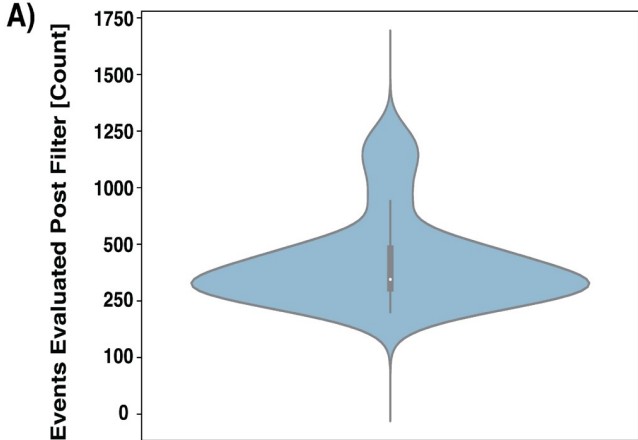
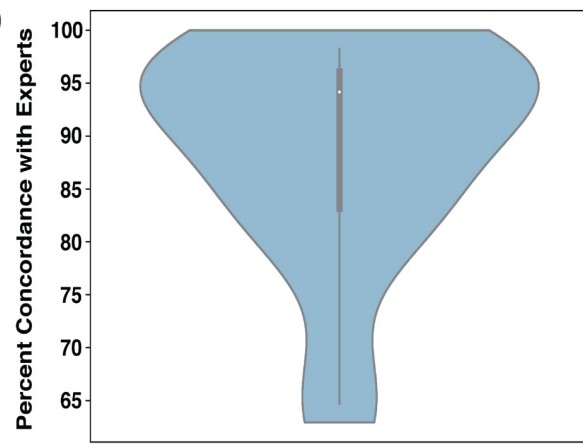

**Fig 3. Responses from curators who evaluated over 648 events.** A) Distribution of the number of events evaluated after filtering survey responses. B) Concordance of responses from curators that evaluated over 648 events with expert consensus genotype labels. **Plot key**: Density plot (blue) represents the overall distribution of the data. Box plot (center of density plot): the thick bar represents the interquartile range of the data; the thin lines extending from the center bar represents the upper and lower adjacent values (i.e., Lower Limit: Q1 − 1.5 × IQR. Upper Limit: Q3 + 1.5 × IQR) in the data. White dot: represents the data median.

Because many curators were anonymous, we screened curators for downstream use based on their concordance with the 'expert' consensus label for the 541 events (Fig 4). Responses were placed into two groups of "top curators": 26 (out of 61) curators above Threshold 1 (90.9% or greater concordance, at least as concordant as the expert with the second lowest concordance), and 37 curators above Threshold 2 (77.7% or greater concordance, at least as concordant as the expert with the lowest concordance–see (S1 Table). We filtered 133 out of 1295 sites because the consensus label of curators above Threshold 1 was different from the consensus label of curators above Threshold 2. 1162 events (527 deletions and 635 insertions) were retained (S4 Fig).

The responses from Threshold 1 and Threshold 2 top curators were generally concordant for homozygous reference, heterozygous, and homozygous variants, with >80% concordance among curators for most variants (Fig 5). Threshold 1 top curators were more concordant than Threshold 2 top curators, particularly for insertions, complex variants, and inaccurate variants (Another_Var), but fewer Threshold 1 curators agreed on the assigned labels. Complex events were particularly challenging, having the lowest levels of concordance for top curators within both groups with a mean concordance of 64% and 47% within top curators above Threshold 1 and 2, respectively.

## Label evaluation

To evaluate the reliability of the top curators' labels for the 527 deletions and 635 insertions, they were compared to the GIAB v0.6 sequence-resolved SV calls and benchmark regions for the Ashkenazi Jewish Trio son[8]. Of the 698 curated sites inside the v0.6 benchmark regions, 669 (94.5%) of the labels assigned by the top curators were concordant with the v0.6 genotype labels (Fig 6).

**Concordance Percent**: High (80% or more concordance); Medium (60–80% concordance); Low (60% or less concordance). **Concordance Count**: High (5 or more curators agreed on the final label); Medium(3–4 curators agreed on the final label); Low(3 or fewer curators agreed on the final label assigned).

The focus of the v0.6 sequence-resolved SV calls was on variants greater than 50bp in size, but we included the filtered v0.6 calls 20 to 49 bp in size in this comparison to help evaluate the reliability of top curators' labels in this size range. 10 of the 29 events discordant between the curators and v0.6 were 20 to 49 bp, and all but one of these appeared to be accurately labeled by curators or could be labeled in multiple ways. For instance, the event could be complex or could contain two or more insertions of different sizes at the same loci. The v0.6 benchmark regions were designed to exclude complex events (i.e., regions with two or more SVs within 1000bp). 11 of 29 discordant events were labeled as complex variants by the top curators (2 of which were also 20 to 49bp in size). S5 Fig includes two examples of these events that were difficult to evaluate by the curators as shown by having 50% or less concordance amongst curators. Upon further curation, all but one or two of these appeared to be true complex variants. Of the remaining 10 discordant events, most appeared to be correctly classified by top curators. However, 2 events were classified as homozygous reference by curators even though another variant was in the same tandem repeat outside the IGV view displayed to curators. This difficulty in accurately classifying complex events in tandem repeat regions highlights the importance of expanding the view to display the entire tandem repeat region for variants overlapping them. Many of the differences between v0.6 and top curators were related to challenges in translating the v0.6 benchmark calls and regions into labels for the curated events. For example, because v0.6 focused on variants >49bp, v0.6 labels were often different if curators labeled a complex variant in which part of the variant

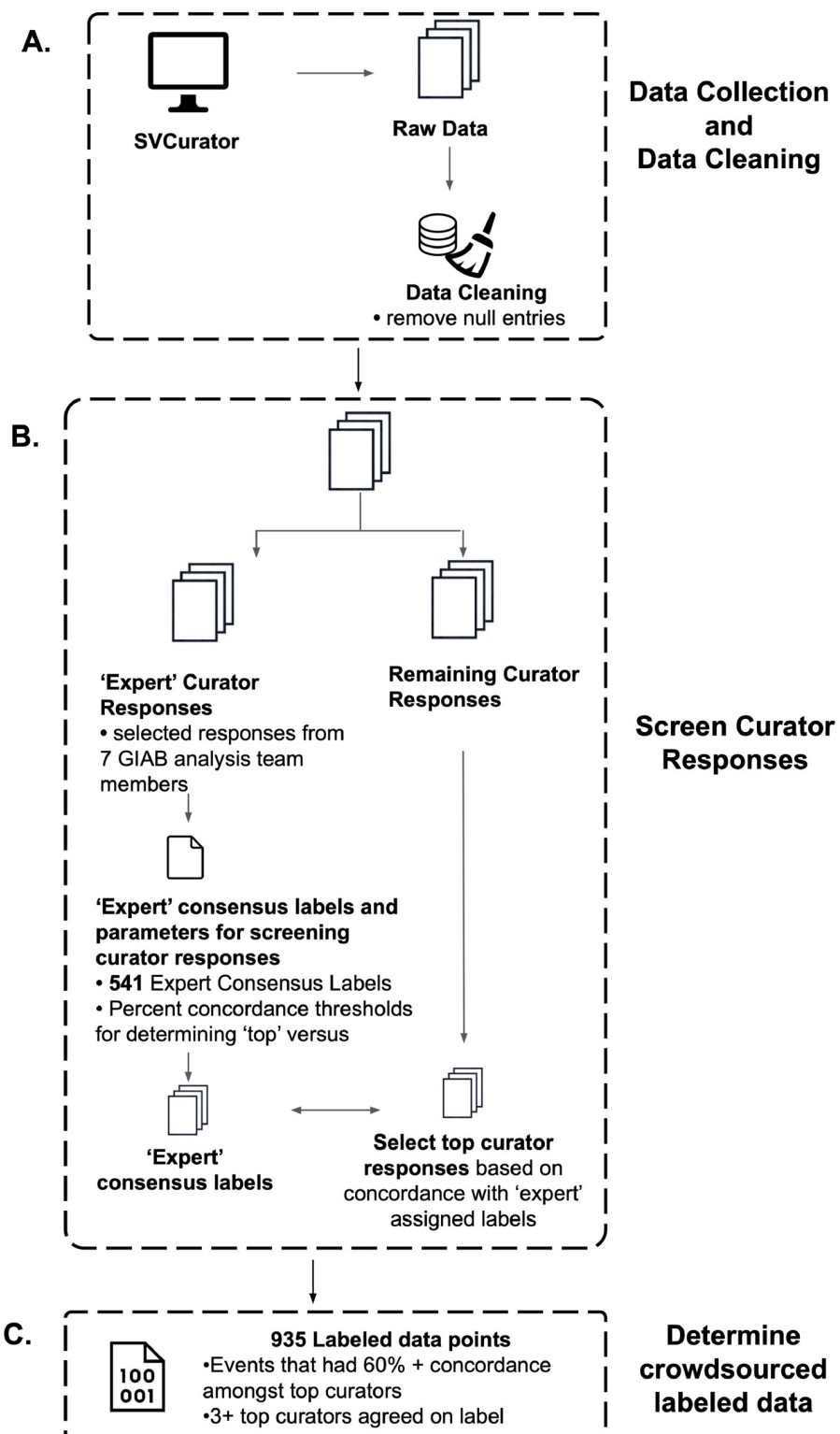

**Fig 4. Schematic summarizing how SVCurator responses were processed to determine the final label for each event.** A) Data Collection and Data Cleaning: Curators evaluated the 1295 events within SVCurator. After removing events that received a low confidence score for genotype assigned and an 'unsure' response for whether an event exists at a particular site, 1273 event remained for analysis. B) Screen Curator Responses: To determine the curator responses that were used to find final labels for the SVCurator events, first consensus labels assigned by 7 'expert' curators were

determined. These 7 'expert' curators were members of the Genome in a Bottle (GIAB) analysis team. Of the 1273 events, 541 were assigned a consensus label by the 'expert' curators, where each event had 68% or greater concordance on the assigned label and 4 or more experts that agreed on the assigned label. Using a leave-one-out strategy, a percent concordance score was found for each 'expert' curator, and the two lowest percent concordance scores (90.9% and 77.7%) were used as a threshold for screening top curators. To find the top curators, labels assigned by each curator were compared to the 541 events and percent concordance with experts was found for each curator. Curators that had 90.9% or greater concordance and 77.7% or greater concordance were considered top curators and their responses were placed in two threshold groups. The responses for these curators were used to find final labels for the SVCurator events. C) Determine crowdsourced labeled data: There were 935 events that were assigned final labels by top curators. These events had at least 60% concordance amongst top curators and at least 3 top curators that agreed on the final label assigned.

was <50bp. There were also cases where multiple nearby variants could be combined into a single variant or separated into multiple variants. Fig 6 summarizes characteristics of the calls discordant between v0.6 and top curators.

We performed additional manual curation to determine the threshold for percent concordance to be included in the final crowdsourced labels. In this re-curation by the authors, we used a dynamic IGV session with new data types, including Oxford Nanopore ultralong reads, in addition to the images displayed in SVCurator. 84 events that were assigned labels with less

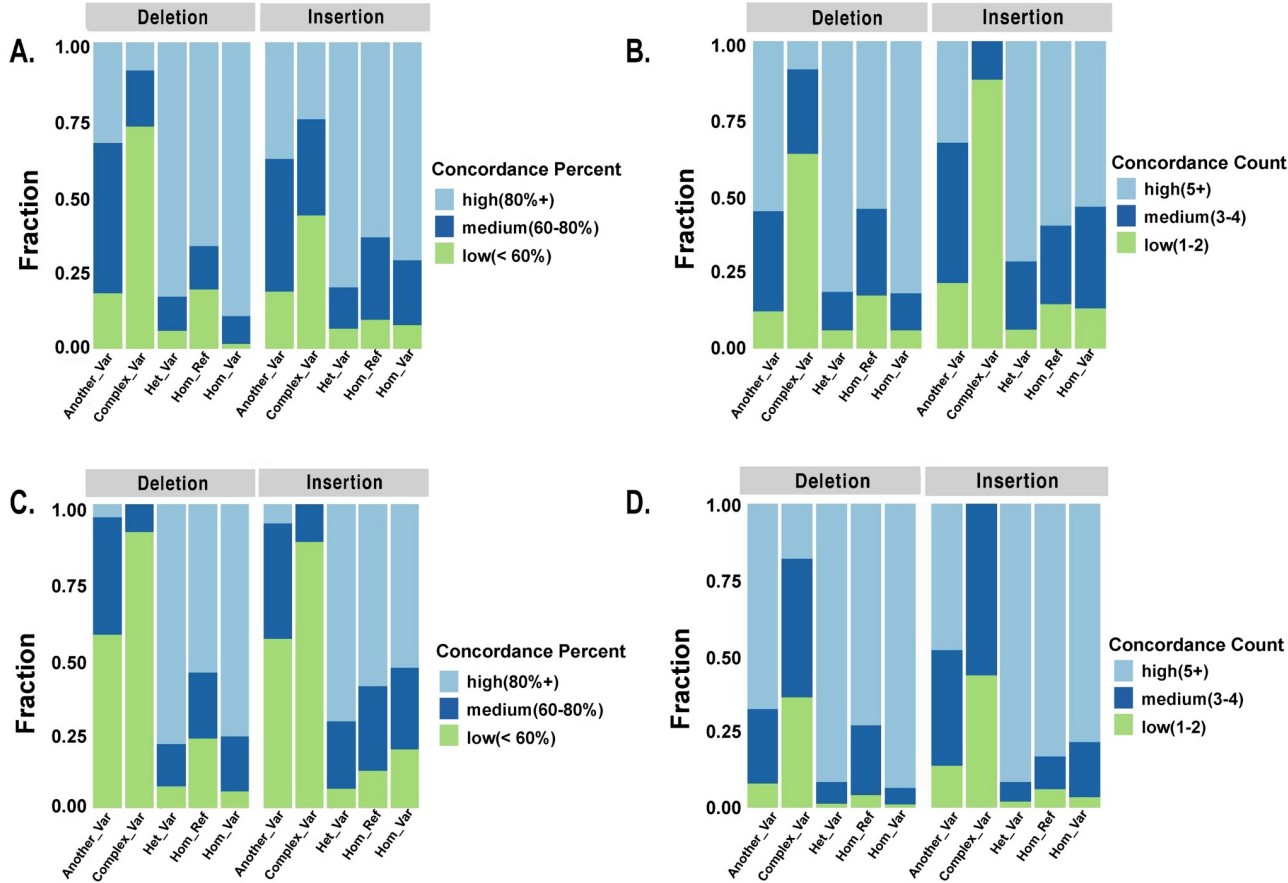

**Fig 5. Concordance evaluation of labels assigned to SVCurator calls by top curators.** A) Percent concordance amongst Threshold 1 top curators on assigned label. B) Fraction of top curators within Threshold 1 that agreed on the assigned label. C) Percent concordance amongst Threshold 2 top curators on assigned label. D) Fraction of Threshold 2 curators that agreed on the assigned label. **Concordance_Percent**: High (80% or more concordance); Medium (60–80% concordance); Low (60% or less concordance). **Concordance_Count**: High (5 or more curators agreed on the final label); Medium (3–4 curators agreed on the final label); Low (3 or fewer curators agreed on the final label assigned).

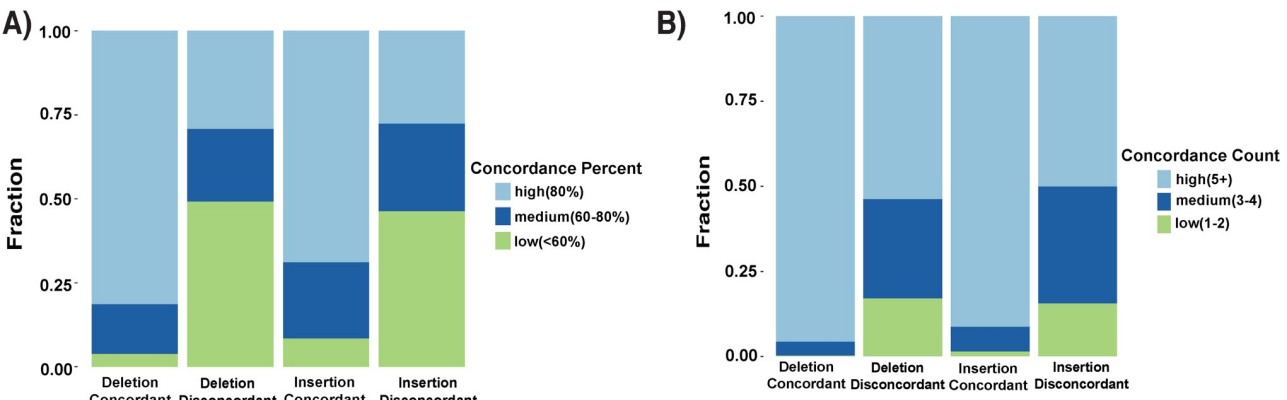

**Fig 6. Comparison of deletions and insertions where SVCurator labels assigned by top curators were either concordant or discordant with v0.6 GIAB benchmark genotypes.** (A) The fraction of calls with high, medium, and low percent concordance among top curators. (B) The fraction of calls with high, medium, and low counts of top curators agreeing on the assigned label.

than 50% concordance amongst all top curators were not included as final labels, because they were highly enriched in calls discordant with v0.6 and were mostly inaccurate or unclear. Upon manual inspection of 44 sites with only 50–60% concordance amongst all top curators, 61% of the events were assigned the correct label. Many of the incorrectly labeled events were not correctly classified as complex variants. Upon manual inspection of 28 sites with 60–70% concordance amongst all top curators, 85% of the events were assigned the correct label. Therefore, only events that were assigned labels with greater than 60% concordance amongst all top curators and at least 3 top curators agreed on the label were included in the final labeled callset. These sites included 496 insertions and 439 deletions with 94% of the events receiving labels of Homozygous Reference, Heterozygous Variant, or Homozygous Variant (Fig 7). Fewer events were homozygous reference than heterozygous or homozygous variant because the events were randomly selected from SV callsets for the son-father-mother trio, and most calls were true SVs in the son. Some events were homozygous reference either because they were FP calls or because they were SVs in the parents that were not inherited by the son, which are more useful controls than randomly generated SVs in the genome. Future evaluations could get a better balance between homozygous reference and true SVs by including more filtered calls from input callsets and by including calls from unrelated individuals.

We also used svviz2 to evaluate the curators' final labeled callset, including variants outside the v0.6 benchmark regions. svviz2 determines whether each read more closely matches the reference allele or the alternate allele or if it is ambiguous. svviz2 generates genotypes [Homozygous Reference, Heterozygous Variant, Homozygous Variant] based on the number of reads that align to the reference and alternate alleles, weighted by their mapping quality scores. We generated svviz2 genotypes from 5 datasets [Illumina 250bp paired end sequencing, Illumina 150bp paired end sequencing, Illumina mate-pair, haplotype-partitioned PacBio and haplotype-partitioned 10x Genomics], and genotypes from each technology were compared to the 879 SVCurator crowdsourced labels that were Homozygous Reference, Heterozygous Variant, or Homozygous Variant. For PacBio and 10x Genomics, calls where either haplotype was heterozygous were conservatively not counted as supporting the label, since almost all the reads on each haplotype should either support the SV or reference allele.

The crowdsourced label for 91% (804 of the 879 events) were supported by at least 2 technologies (Fig 8). Upon detailed manual curation with new long read data in a dynamic IGV session, the curators' genotype was correct for 7 of the 11 events with no svviz genotype

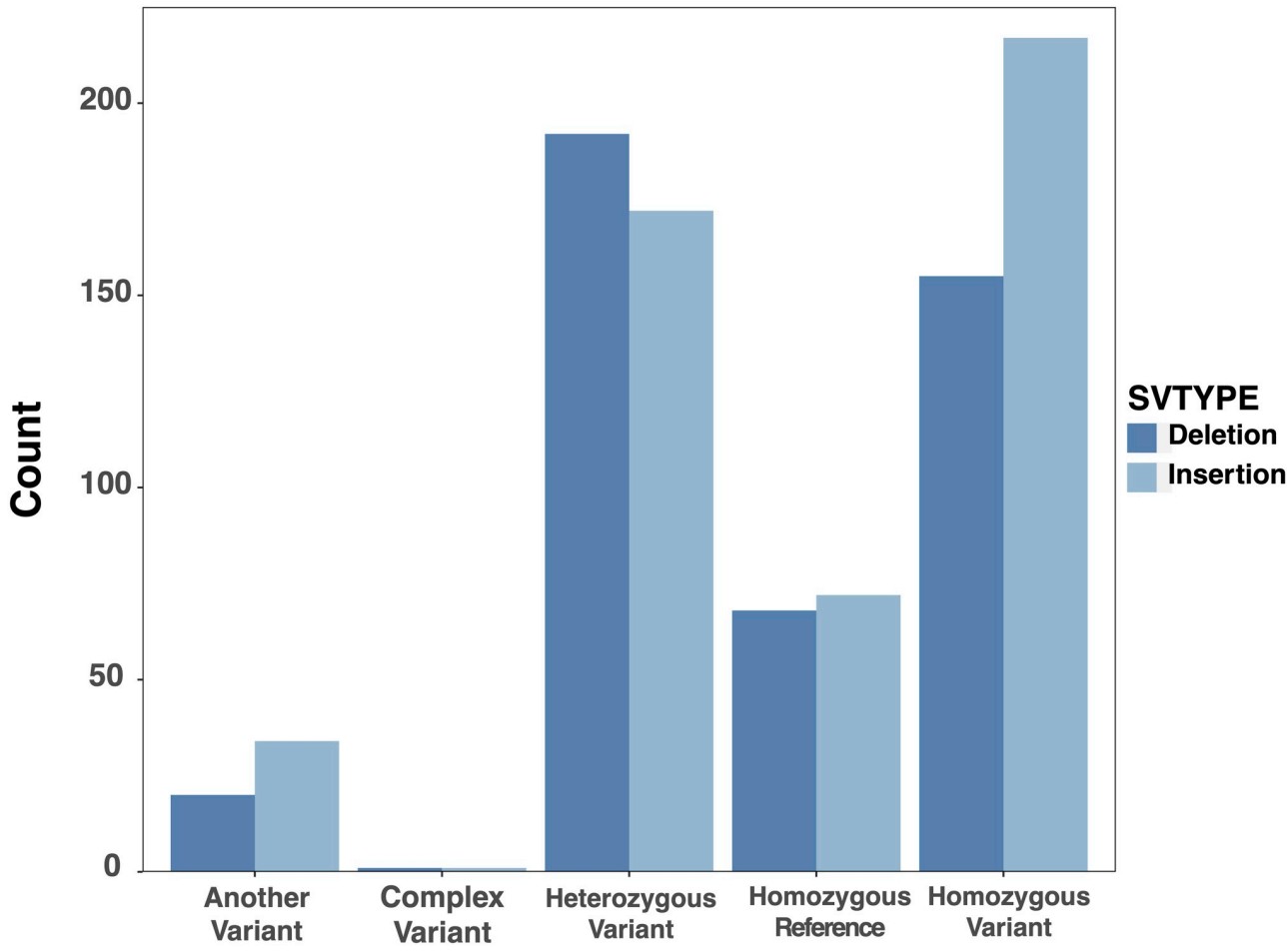

**Fig 7. A summary of the final crowdsourced SVCurator labels.**

support, and the svviz genotypes did not agree mostly due to slight inaccuracy in the SV breakpoints or size. The events incorrectly classified by the curators were mostly complex variants that the curators determined to be simple heterozygous or homozygous variants. There were also 64 events where only 1 technology supported the crowdsourced label, 40 only by PacBio and 16 only by mate-pair (S2 Table). We performed detailed manual curation of 10 of the sites supported only by PacBio, and the curators were correct in 8 of them, called one event a simple deletion when in fact it was complex due to a 500bp insertion on the other haplotype, and called one event homozygous variant even though it was slightly larger than the predicted size range. We performed detailed manual curation of 10 of the sites supported only by Illumina mate-pair, and the curators were correct for 4 of them, 3 should have been called complex due to different variants on opposite haplotypes, and 2 were unclear due to alpha satellites near the centromere, and 1 was likely a deletion in a larger duplicated region. Upon curation of all 6 sites only supported by 250bp illumina reads, 5 were correctly called 20-100bp indels in TRs between the 150bp and 250bp illumina read lengths, and one should have been called Another_Var since it was larger, though this was not obvious in the static images displayed in SVCurator. These results further support the accuracy of the crowdsourced labeled events, including those outside the v0.6 benchmark regions, and also points to causes of the small

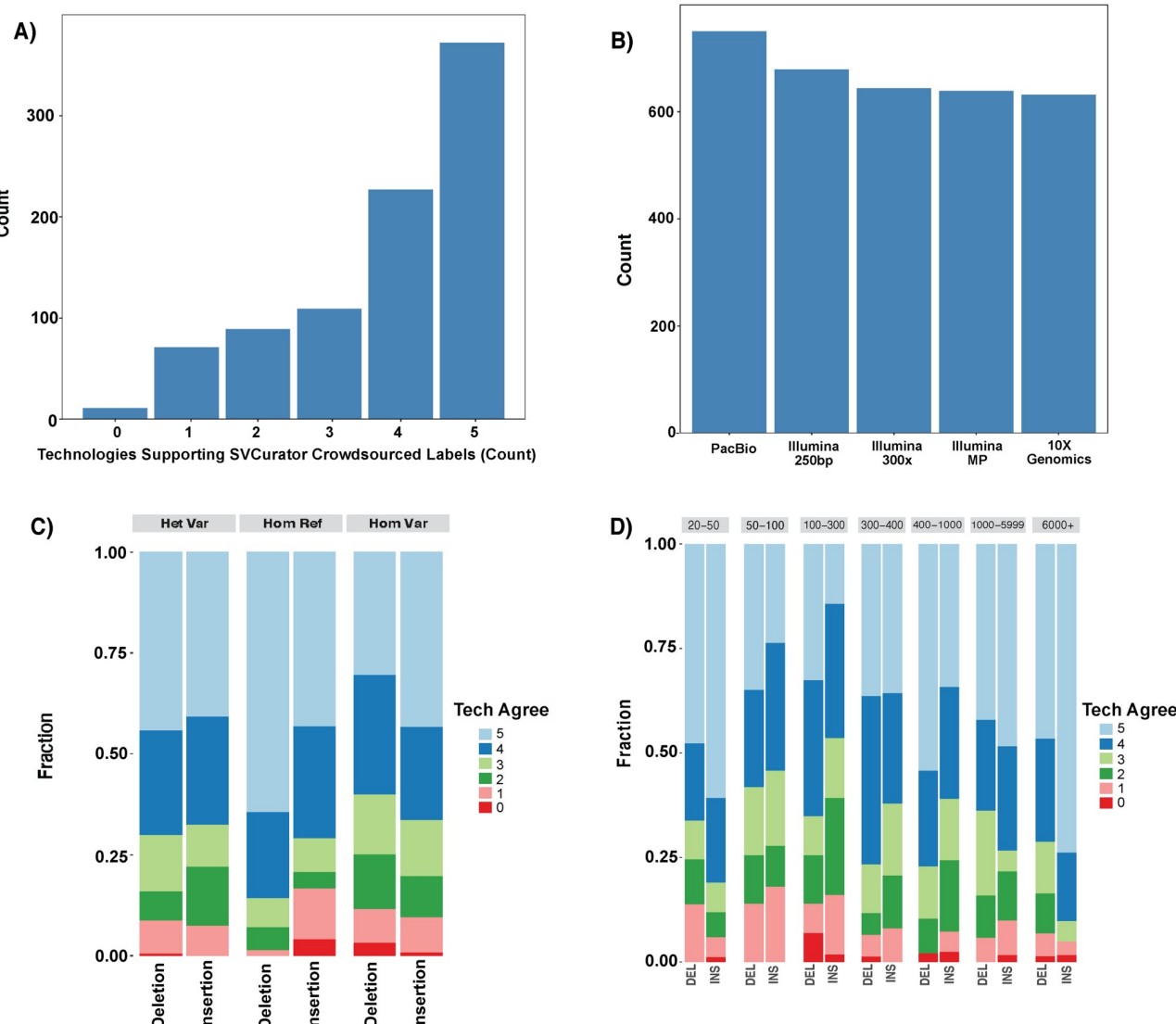

**Fig 8. svviz2 genotypes support the 879 SVCurator crowdsourced labels.** A) A summary of the number of technologies whose svviz2 genotypes support the SVCurator genotype label. 92.2% of the events were supported by at least 2 technologies. B) A count of the number of genotypes from each technology that match the SVCurator crowdsourced labels. C) A summary of the number of technologies that had genotype scores supporting the crowdsourced label as summarized based on label and variant type; and, D) by size of the event.

number of curators' errors due to complex variants or imprecisely called variants in repetitive regions of the genome.

## Discussion

We performed a crowdsourcing experiment with a new tool SVCurator, which incorporated different visualizations of short, long and linked read sequencing data to enable users to evaluate SV calls. For the first time, curators could evaluate multiple sources of evidence for each call in one app interface. We displayed svviz2 images of reads from 3 different Illumina sequencing methods, haplotype-partitioned PacBio, and haplotype-partitioned 10x Genomics aligned to reference and alternate alleles, as well as dotplots to visualize repetitive regions. The app also included IGV images that display Illumina 250bp, PacBio and 10x

Genomics reads. Both the IGV and svviz2 images included annotations of repetitive regions. These features enabled participants to more easily evaluate deletions and insertions, including repetitive regions.

This study suggests that a group of participants can accurately curate SV calls by evaluating a variety of static images from multiple sequencing technologies. While all of the metrics are specific to this group of curators, we learned several things that could improve future curation studies. In general, isolated heterozygous and homozygous variants and homozygous reference regions were accurately labeled as long as the variant size and type were accurate. A major cause of lower concordance scores were complex events, which were often located in repetitive regions of the genome. One limitation of this experiment was that the IGV images sometimes only displayed part of a large tandem repeat and missed parts of a complex variant, so we recommend that future curation images display a larger region fully encompassing any tandem repeats overlapping the SV. Crowdsourcing studies specifically focused on complex SVs should also likely provide tutorials with several examples of complex events, including where more than one SV is in *cis* and in *trans*. In addition, SVs that had inaccurate breakpoints or size estimates were inconsistently classified as Another_Var by curators, particularly confusing curators with less expertise. Ensuring IGV images encompass entire tandem repeats also should help ensure more accurate curations of inaccurately defined SVs. Enabling curators to dynamically zoom in and out of IGV displays would also make curation of more challenging events like those in tandem repeats or segmental duplications. Dynamic visualizations could also enable users to scroll through higher depth of reads. For future studies that may not have as extensive data for curation, we recommend including long reads if curation of SVs in tandem repeats is important and including images from svviz if curation of insertions is important.

The crowdsourced labels derived from this study will be useful training datasets for machine learning studies that evaluate SVs, and could be used as a resource to improve SV calling methods. The calls could also be used as a resource to help members of the clinical genomics community improve their evaluation of SVs. Crowdsourcing could also yield more reliable resources that could improve clinical interpretations of SVs as many of the guidelines are qualitative[11]. Finally, this study demonstrates that crowdsourcing is a useful strategy for evaluating SV calls and the results of crowdsourcing could yield results that may be useful in improving SV tools and analyses approaches in multiple domains.

## Materials & methods

### SVCurator events

SVs and large indels from the Ashkenazi Jewish Trio son (NA24385/HG002) were randomly sampled from the Genome In A Bottle (GIAB) union callset [union_171212_refalt.sort. vcf]. The calls and distance metrics can be found at: ftp://ftp-trace.ncbi.nlm.nih.gov/ReferenceSamples/giab/data/AshkenazimTrio/analysis/NIST_UnionSVs_12122017/. The calls are a union set of 1+ million sequence-resolved calls ≥20bp from 5 short, long, and linked read technologies and over 30 SV discovery methods. The events were randomly sampled to obtain equal numbers of events in the following size bins: 20-49bp, 50-100bp, 100-300bp, 300-400bp, 400-1000bp, 1000-5999bp, 6000+ bp. 579 putative deletions and 716 putative insertions were included in the app.

### Participant recruitment

Participants were recruited from the Genome in a Bottle Analysis Team (https://groups.google.com/forum/#!forum/giab-analysis-team) and from the genomics community via the

@GenomeinaBottle Twitter account. SVCurator was made available to the public for one month to allow participants to evaluate the events within the app. An incentive of co-authorship on the current publication was offered for participants who curated at least half of the events (648 or more events).

### SVCurator app interface

SVCurator (www.svcurator.com) is a Python Flask-based app (Fig 1) and uses SQLite3 as a database management system. User login was implemented using Google OAuth 2.0. The SVCurator app was deployed using pythonanywhere[www.pythonanywhere.com].

*App Images*: The interface consisted of four thumbnail images for each event and a set of four questions. The four thumbnail images consisted of the following: IGV image, svviz2 PacBio haplotype-partitioned read aligned image, svviz2 10X Genomics haplotype-partitioned read aligned image, svviz2 dotplot image representing reference versus alternate allele. A lightbox contained additional images to describe each event, and included the following: svviz2 read aligned image for haplotype and non-haplotype-partitioned PacBio data, 10X Genomics haplotype-partitioned data, Illumina 6kb Mate Pair data, Illumina HiSeq 250bp read length data, Illumina HiSeq 300x read depth data; svviz2 dotplots: represent reads with highest mapping quality versus reference and alternate allele, reference allele versus alternate allele, reference allele versus reference allele, and alternate allele versus alternate allele. Images included in the lightbox allowed curators to zoom in on sections of the SV call that required a more close evaluation. Each curator evaluated the same events for the first 43 events, and events 44–1295 were randomized for each user.

*Questions*: Participants were given the structural variant call: unique ID, size, chromosome number, start and end coordinates. For each event, curators evaluated the putative SV type, determined whether an event exists within 20% the size of the variant, and the genotype for each event. The questions included in SVCurator were designed to describe the size accuracy and genotype of each SV call. Members of the GIAB community helped structure and finalize the questions included in the app. Curators described each event by responding to the following questions:

Question 1: Does a [insertion/deletion] exist at this site that has a size between [start coordinate] bp and [end coordinate]bp?

- Yes [Answer Q3 and Q4]

- No [Answer Q2 only]

- Unsure [move on to next variant]

Question 2: Does any other variant exist at this site?

- Yes

- No

Question 3: Select the genotype for this variant.

- Homozygous Reference

- Heterozygous Variant

- Homozygous Variant

- Complex [ie: 2+ variants in this region] or difficult

- Unsure

Question 4: Note confidence selected in the genotype

- 2 [most confident]

- 1

- 0 [least confident]

   Comment Box: included to give curators the opportunity to add additional comments to describe each event or report any user interface issues (ie: images that may have not rendered properly).

   Responses were collected over the course of one month after the app was made publically available. Participants were also provided with a tutorial that describes general guidelines for analyzing SV calls (https://lesleymaraina.github.io/svcurator_tutorial_2/).

### Data used to generate images

Aligned reads for the Ashkenazi Jewish Trio son (NA24385/HG002) were used to generate the images used within the app. The BAM files are publicly available from the GIAB FTP site as listed in Table 1.

### svviz2 images

svviz2 (version 2.0a3, https://github.com/nspies/svviz2) aligned-read images and dotplots were generated for each event. Svviz2 is a SV visualization tool that identifies reads that support a reference allele, alternate allele (supports a SV call), or are ambiguous. 10X Genomics (10X) and PacBio sequencing images were haplotype-partitioned, PacBio reads were haplotype-partitioned using WhatsHap and reads for both 10X and PacBio were subsequently aligned to a reference and alternate allele using svviz2.

### IGV images

Integrative Genomics Viewer (IGV) version 2.4.6 was also used to generate images for each event and displays tracks representing reads from haplotype-partitioned PacBio and 10X

**Table 1. Summary of data used to generate images within SVCurator.**

| Sequencing Technology | Library Type | Average Read Length [bp] | Average Read Depth | BAM File Links |
|---|---|---|---|---|
| Pacific Bioscience (PacBio) | Whole Genome Sequencing (WGS) Single End | 10-11Kb | 69x | ftp://ftp-trace.ncbi.nlm.nih.gov/ReferenceSamples/giab/data/AshkenazimTrio/HG002_NA24385_son/PacBio_MtSinai_NIST/Baylor_NGMLR_bam_GRCh37/HG002_PB_70x_RG_HP10XtrioRTG.bam |
| 10X Genomics Chromium Sequencing | WGS Linked Reads | 2x98 | 50x | ftp://ftp-trace.ncbi.nlm.nih.gov/ReferenceSamples/giab/data/AshkenazimTrio/analysis/10XGenomics_ChromiumGenome_LongRanger2.2_Supernova2.0.1_04122018/GRCh37/NA24385_300G/HG002_10x_84x_RG_HP10xtrioRTG.bam |
| Illumina HiSeq 2500 | WGS mate-pair | 2x100 [6000bp insert size] | 13-14x | ftp://ftp-trace.ncbi.nlm.nih.gov/ReferenceSamples/giab/data/AshkenazimTrio/HG002_NA24385_son/NIST_Stanford_Illumina_6kb_matepair/ |
| Illumina HiSeq 2500 | WGS paired-end | 2x250 | 40-50x | ftp://ftp-trace.ncbi.nlm.nih.gov/ReferenceSamples/giab/data/AshkenazimTrio/HG002_NA24385_son/NIST_HiSeq_HG002_Homogeneity-10953946/NHGRI_Illumina300X_AJtrio_novoalign_bams/ |
| Illumina HiSeq 2500 | WGS paired-end | 2x148 | 296.83x | ftp://ftp-trace.ncbi.nlm.nih.gov/ReferenceSamples/giab/data/AshkenazimTrio/HG002_NA24385_son/NIST_HiSeq_HG002_Homogeneity-10953946/NHGRI_Illumina300X_AJtrio_novoalign_bams/ |

Genomics Chromium, Illumina Paired-End reads (250 base pair read length), and tracks describing repetitive regions of the genome (S1 Fig). Soft-clipping was shown by coloring ends of clipped reads but clipped bases were not shown in order to make the view cleaner. Within each IGV image, the putative insertion or deletion was displayed along with flanking regions. For deletions, the flanking regions that were 20% of the size of the variant at the start and end position of each displayed event, and for insertions the flanking regions were 1.6 times the size of the variant at the start site and the region flanking the end position of the event was 70% of the size of the event.

## Crowdsourced labels

Each event was assigned one of the following genotype labels:

- Homozygous Reference [Hom_Ref]

- Heterozygous Variant [Het_Var]

- Homozygous Variant [Hom_Var]

- Complex [ie: 2+ variants in this region] or difficult [Complex_Var]

- Another variant exists at this site/A variant more than 20% different in size exists at this site [Another_Var]

  Responses were processed as described in Table 2 to generate the labels above.
  Responses were initially filtered as follows, if one of the following was true, responses were not included in the label assessment:

- curator provided no response to the questions

- curator selected 'Unsure' for the question 1

- curator selected 'least confident' for confidence in the genotype selected

To determine the curator responses that would be used to generate final labels for each event, curator responses were screened based on concordance with 'expert' consensus labels (Fig 8) since there are currently no comprehensive ground truth labels available for these events. Seven 'expert' curators were identified from the GIAB Analysis Team based on their known prior experience curating SVs. Each event was assigned one of the following labels ('Hom_Ref' [Homozygous Reference], 'Het_Var' [Heterozygous Variant], 'Hom_Var' [Homozygous Variant], 'Another_Var' [Another Variant Exists within 20% of the size of the variant],

**Table 2. Description of genotype label assignment based on responses to survey questions.**

| Genotype Label | Question 1: Does a [insertion/deletion] exist at this site that has a size between [start coordinate] bp and [end coordinate]bp? | Question 2: Does any other variant exist at this site? | Question 3: Select the genotype for this variant. | Question 4: Note confidence selected in the genotype |
|---|---|---|---|---|
| Homozygous Reference | No | No | - - - | - - - |
| Heterozygous Variant | Yes | - - - | Heterozygous Variant | 1 + |
| Homozygous Variant | Yes | - - - | Homozygous Variant | 1+ |
| Another Variant Exists | No | Yes | - - - | - - - |
| Complex | Yes | - - - | Complex | 1+ |

'Complex_Var' [Complex Event]). The number of 'expert' curators that agreed on a label was determined as well as the percent concordance between 'experts' for each event. The percent concordance was determined based on the ratio of the number of 'expert' curators that agreed on a label versus the total number of 'expert' curators that evaluated each event. Consensus labels were assigned based on majority vote and events used for screening curators were those that had at least 3 expert curators agree on a label with at least 67% concordance. A leave-one-out strategy was used to determine the level of concordance between 'expert' curators. Two thresholds were set to determine remaining curators whos evaluations would be used for assigning crowdsourced labels. These thresholds were set based on the two lowest concordance levels between 'expert' curators.

Responses from curators with 77.7% concordance with experts and 90.9% concordance with 'expert' curators were included for further analysis. Only the events that had concordant labels between curators in the two threshold groups were used for further label analysis. Labels were determined for these events and events with at least 50% concordance amongst all top curators were evaluated further. Select events were manually inspected, and it was determined that sites with 60% or greater concordance with at least 3 curators that agreed on the label were included in the final labeled dataset.

## SVCurator label corroboration—GIAB v0.6 high confidence call genotype labels and svviz2 genotype labels

**v0.6 Genotype Labels.** The GIAB v0.6 Benchmark SV Set [10] was generated using the following heuristics from the same union vcf sampled for the SVCurator variants above, which came from 30 callers and 5 technologies on all three members of the GIAB Ashkenazi trio at ftp://ftp-trace.ncbi.nlm.nih.gov/ReferenceSamples/giab/data/AshkenazimTrio/analysis/NIST_UnionSVs_12122017/union_171212_refalt.sort.vcf.gz

1. Sequence-resolved variants with at least 20% sequence similarity were merged into a single vcf line using SVanalyzer (https://github.com/nhansen/SVanalyzer)

2. Variants supported by at least two technologies (including BioNano and Nabsys) or by at least 5 callsets from a single technology had evidence for them evaluated and were genotyped using svviz2 with the four datasets in Table 3. Genotypes from Illumina and 10x were ignored in tandem repeats >100bp in length, and genotypes from PacBio were ignored in tandem repeats >10000bp. Genotypes from all datasets were ignored in segmental duplications >10000bp. If the genotypes from all remaining datasets were concordant, and PacBio supported a genotype of heterozygous or homozygous variant, then the variant was included in downstream analyses.

3. If two or more supported variants ≥50bp were within 1000bp of each other, they were filtered because they are potentially complex or inaccurate.

In addition, benchmark regions were formed with the following process:

1. Call variants from 3 PacBio-based and 1 10x-based assemblies

2. Compare variants from each assembly to our v0.5.0 PASS calls for HG2 allowing them to be up to 20% different in all 3 distance measures, and only keep variants not matching a v0.5.0 call.

   a. Cluster the remaining variants from all assemblies and keep any that are supported by at least one long read assembly

**Table 3. Heuristics used to determine HG002 genotypes.**

| Technology | Minimum ALT and REF count [ALT+REF$\geq$n] | ALT and REF Count Ratio (x = ALT/(REF+ALT)) | Genotype Label |
|---|---|---|---|
| PacBio | n$\geq$8 | x<0.1 | Homozygous Reference (GT = 0/0) |
| | | 0.25<x<0.75 | Heterozygous Variant (GT = 0/1) |
| | | x>0.9 | Homozygous Variant (GT = 1/1) |
| Illumina 250bp and MP Illumina | n$\geq$8 | x<0.05 | Homozygous Reference (GT = 0/0) |
| | | 0.1<x<0.9 | Heterozygous Variant (GT = 0/1) |
| | | x>0.95 | Homozygous Variant (GT = 1/1) |
| 10x Genomics | n$\geq$5 | x1<0.05 AND x2<0.05 | Homozygous Reference (GT = 0/0) |
| | | (x1>0.95 AND x2<0.05) OR (x2>0.95 AND x1<0.05) | Heterozygous Variant (GT = 0/1) |
| | | x1>0.95 AND x2>0.95 | Homozygous Variant (GT = 1/1) |

Genotypes for HG002 were determined using a heuristics based strategy by determining cut-offs for weighted alternate and reference(REF) counts [ALT/(REF+ALT)]. The cut-offs were determined manually from looking at distributions for different size ranges. For each technology, a minimum ALT and REF count was set and genotypes were determined based on the ratio of REF to ALT counts. Variants that did not meet the criteria in this table were not included in the v0.6 comparison.

3. Find regions for each assembly that are covered by exactly one contig for each haplotype.

4. Find the number of assemblies for which both haplotypes cover each region

5. Subtract regions around variants remaining after #2, using svwiden's repeat-expanded coordinates, and expanded further to include any overlapping repetitive regions from Tandem Repeat Finder, RepeatMasker SimpleRepeats, and RepeatMasker LowComplexity, plus 50bp on each end.

6. High confidence regions are regions in #4 covered by at least 1 assembly minus the regions in #5.

7. Further exclude any regions in the Tier 2 bed file of unresolved and clusters of variants, unless the Tier 2 region overlaps a Tier 1 PASS call.

## Supporting information

**S1 Fig. Images were generated for each event from Integrated Genome Viewer, svviz2.** A putative 467bp deletion is shown. Svviz2 generates read aligned images for each short read and long read sequencing technology. A) svviz2 read aligned image - 10x Genomics (read length = 98bp; read depth = 50x). Reads were aligned to reference and alternate allele by svviz2. B) svviz2 read aligned image—Illumina HiSeq (read length = 250 bps; read depth = 40-50x). C) svviz2 read aligned image—Illumina HiSeq (read length = 148 bps; read depth = 296.83x). D) svviz2 read aligned image—Illumina Mate Pair (read length = 100 bps; insert size = 6000bp; read depth = 13-14x). E) svviz2 read aligned image—Haplotype separated PacBio (read length = 10-11kb; read depth = 69x). Reads were haplotype separated using WhatsHap and aligned to reference and alternate allele by svviz2. F) Svviz2 dotplot displaying

reference versus alternate allele. G) Svviz2 dotplot PacBio read with the highest mapping quality score versus the alternate allele. H) IGV image showing reads aligned to a putative variant. IGV tracks include: Haplotype separated PacBio reads, Haplotype Separated 10x Genomics reads, Illumina 250x250bp paired end sequencing, and tracks to describe repeat regions (low complexity repeats and segmental duplications). All reads were aligned to GRCh37 human reference genome. I) Svviz2 dotplot displaying a putative deletion in a tandem repeat.
(TIF)

**S2 Fig. Summary of the number of curations for each event.** 61 curators evaluated SVCurator events. Each of the 1295 sites were curated on average 11 times with 1290 events curated at least 3 times.
(TIF)

**S3 Fig. An evaluation of the time to curate each SVCurator event.** A) Overall distribution of the average time to curate events. B) Distribution of the average time to curate each event for each curator.
(TIF)

**S4 Fig. Evaluation of concordance between Threshold 1 Top Curators (curators that had at least 90.9% concordance with experts) and Threshold 2 Top Curators (curators that had at least 77.7% concordance with experts).**
(TIF)

**S5 Fig. Examples of A) a small SV call and B) a large SV call that were discordant between consensus labels assigned by curators and the v0.6 high confidence genotypes discordant sites.** IGV images showing examples of two events that had less than 50% concordance for the label assigned by the curators.
(TIF)

**S1 Table. Concordance scores amongst 'expert' curators.**
(TIF)

**S2 Table. List of curated sites with 0 or 1 concordant genotypes using svviz, including results of manual re-curation of selected sites.**
(XLSX)

**S1 Data. SVCurator final labels (>60% curator concordance with at least 3 curators agreeing on the final label).**
(XLSX)

**S2 Data. SVCurator final labels with top curator statistics.**
(TSV)

**S3 Data. SVCurator final labels with labels assigned by each curator.**
(TSV)

**S4 Data. List of sequencing technologies and variant callers used to discover the SV calls within SVCurator.**
(XLSX)

## Acknowledgments

We thank many Genome in a Bottle Consortium Analysis Team members for helpful discussions about design of the SVCurator app and the experiment. We would especially like to

thank Nancy Hansen of the National Human Genome Research Institute for advice on implementing SVAnalyzer as well as input on interpreting structural variant data.

## Author Contributions

**Conceptualization:** Lesley M. Chapman, Noah Spies, Aaron M. Wenger, Marc Salit, Justin M. Zook.

**Data curation:** Lesley M. Chapman, Noah Spies, Chun Shen Lim, Andrew Carroll, Giuseppe Narzisi, Christopher M. Watson, Christos Proukakis, Wayne E. Clarke, Naoki Nariai, Eric Dawson, Garan Jones, Daniel Blankenberg, Christian Brueffer, Chunlin Xiao, Sree Rohit Raj Kolora, Noah Alexander, Paul Wolujewicz, Azza E. Ahmed, Graeme Smith, Saadlee Shehreen, Aaron M. Wenger, Justin M. Zook.

**Formal analysis:** Lesley M. Chapman.

**Methodology:** Andrew Carroll.

**Project administration:** Justin M. Zook.

**Resources:** Marc Salit, Justin M. Zook.

**Software:** Lesley M. Chapman, Noah Spies, Patrick Pai.

**Supervision:** Marc Salit, Justin M. Zook.

**Visualization:** Lesley M. Chapman, Noah Spies, Andrew Carroll, Justin M. Zook.

**Writing – original draft:** Lesley M. Chapman.

**Writing – review & editing:** Lesley M. Chapman, Justin M. Zook.

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
