## [Decision Letter · Decision Letter 0]

26 Nov 2019

Dear Dr Zook,

Thank you very much for submitting your manuscript 'SVCurator: A Crowdsourcing app to visualize evidence of structural variants for the human genome' for review by PLOS Computational Biology. Your manuscript has been fully evaluated by the PLOS Computational Biology editorial team and in this case also by independent peer reviewers. The reviewers appreciated the attention to an important problem, but raised some substantial concerns about the manuscript as it currently stands. While your manuscript cannot be accepted in its present form, we are willing to consider a revised version in which the issues raised by the reviewers have been adequately addressed. We cannot, of course, promise publication at that time.

Sincerely,

Ilya Ioshikhes

Associate Editor

PLOS Computational Biology

William Noble

Deputy Editor

PLOS Computational Biology

[LINK]

Reviewer's Responses to Questions

**Comments to the Authors:**

Reviewer #1: The authors present a tool called SVCurator, which allows users to view images of sequencing reads from multiple different platforms for the same individual and answer questions about the presence of structural variants. The authors used this tool to interrogate putative SVs for HG002 via crowd sourcing and via expert review. The resulting labels were compared across the expert reviewers, the crowd, and compared to the GIAB benchmark calls.

Major Comments:

SVCurator is a valuable tool for visualizing sequence level evidence for structural variants across sequencing platforms. I think the community will benefit from this tool and the results of the study. I have no major suggestions.

Minor Comments:

- I’m curious about the events that the experts commonly agreed upon or disagreed most on. Can you describe these in terms of their qualitative features? Are they similar to what the non-experts agreed or disagreed upon?

- In Figure 1, Q1 shows a question about the size range for this variant – how are those ranges generated?

- Did you include any controls (e.g. images of regions where no putative SV call existed)?

- Do you have any information about experts and the crowd make their decisions? Do they weight the data in the same ways or look for particular features?

- Were there any individual SVs (e.g. rather than classes of SVs) that the crowd performed better than the experts? If so, it would be interesting to describe these.

- Figure 3 – These may be easier to read as a violin plot.

- Can you explain why the variants in the “another_var” category have such high discordance levels?

- When you say that on page 16 that the events were manually inspected – do you mean by the Authors? Or can you explain how this manual inspection is different than what has been described in other places in the manuscript?

- Figure 9 is really helpful. I think it would help the reader understand the methods and definitions of the various categories if you create something like this to include earlier on (e.g. a Figure 1A).

- Overall I would like to hear more about SVCurator – what parameters or visualizations can be tuned or varied for future studies. I imagine other groups may not have quite so much data on the same sample, but clinical groups, core labs, and SV detection tool developers may be interested in using SVcurator to review data. How flexible will SVCurator be in other studies when not all data types are present? Can you describe the API or general approach for a future user to use this tool?

- Are there any limitations of SVCurator? For example, how much depth can it reasonably display? Does it randomly downsample, etc. Are the images static (e.g. screenshots from IGV) or are users able to use IGV features to scroll, group or color reads, etc?

Reviewer #2: The manuscript entitled “SVCurator: A Crowdsourcing app to visualize evidence of structural variants for the human genome” by Dr. Zook and colleagues described an experience of crowdsourcing manual curation of germline structural variations (SV) of a Genome In A Bottole (GIAB) sample by “recruiting” 61 curators to manually review 1235 putative insertions and deletions. The user interface displays all the relevant information generated by diverse technology together on the same page (e.g. PacBio long read, 10X Genomics reads and Illumina reads), which is useful when evaluating as the user can zoom and scroll pretty easily. Overall, having a review strategy of crowed sourced evaluations to generate a reference SV data is an interesting approach and binning the curators into experts (threshold 1 and non-experts (threshold 2) is a good strategy to avoid reducing the quality of the results due to the input from an inexperienced reviewer.

Major issues:

1) The interface appears to be designed for the specific task of reviewing the GIAB samples. There is no way to upload a custom set of SVs. Without an interface for uploading the custom data, it is unclear how the publication of the tool can benefit research community was. The authors also need to provide tutorial on how to upload raw SV data/images, recruit reviewers, and collect and assess results.

2) The original SVs were generated from 30 variant callers. Can the author indicate the accuracy and sensitivity of each caller based on the curation consensus?

3) In Discussion, the authors mentioned that the crowdsourcing results could be further validated by PCR. Given the long-read sequencing support, would PCR validation still be required? It would be important to present how many of the crowdsourcing results were corroborated by at least one long-read technology (i.e. PacBio, 10X or long-read library). This information was not presented in Figure 8. It is intriguing to see that a small proportion of the SVs were supported only by <=1 technology (Figure 8A). It will be important to explain why these events are considered real.

4) The quality of the manuscript needs to be improved. The text label of Figure 1 is not readable, Figure 7 was of poor quality; Figure 8 used different width to show the bar plot and Figure 9 was first referred on page 15, ahead of Figure 6. A number of paragraphs describe the performance of a specific group of curators working with this specific dataset. This content may need to be removed as it is of little scientific interest and yields little insight. The phrase "python flask web based app" appears throughout the manuscript which does not make sense for people who are not in the field of software development.

Minor issues

1) The “help” tab on the web page is broken, and there is another dead link on the page which raises the concern that an additional image isn’t being rendered. Also, I’m not seeing any that shows where the soft clipped bases are located, which can be important for SV review. They do show realignment, but without checking the softclip orientations and locations, this can be misleading.

**Have all data underlying the figures and results presented in the manuscript been provided?**

Reviewer #1: Yes

Reviewer #2: Yes

PLOS authors have the option to publish the peer review history of their article (what does this mean?). If published, this will include your full peer review and any attached files.

Reviewer #1: No

Reviewer #2: No

---

## [Decision Letter · Decision Letter 1]

7 May 2020

Dear Zook,

We are pleased to inform you that your manuscript 'A crowdsourced set of curated structural variants for the human genome' has been provisionally accepted for publication in PLOS Computational Biology.

Best regards,

Ilya Ioshikhes

Associate Editor

PLOS Computational Biology

William Noble

Deputy Editor

PLOS Computational Biology

Reviewer's Responses to Questions

**Comments to the Authors:**

Reviewer #1: The authors addressed all of my comments; I recommend this manuscript for publication.

Reviewer #2: The authors have addressed all the concerns raised in the previous review.

**Have all data underlying the figures and results presented in the manuscript been provided?**

Reviewer #1: Yes

Reviewer #2: Yes

PLOS authors have the option to publish the peer review history of their article (what does this mean?). If published, this will include your full peer review and any attached files.

Reviewer #1: No

Reviewer #2: No

---

## [Editor Report · Acceptance letter]

9 Jun 2020

PCOMPBIOL-D-19-01608R1 

A crowdsourced set of curated structural variants for the human genome

Dear Dr Zook,

I am pleased to inform you that your manuscript has been formally accepted for publication in PLOS Computational Biology. Your manuscript is now with our production department and you will be notified of the publication date in due course.

With kind regards,

Laura Mallard
